# Curcumin as a Dual Modulator of Pyroptosis: Mechanistic Insights and Therapeutic Potential

**DOI:** 10.3390/ijms26157590

**Published:** 2025-08-06

**Authors:** Dong Oh Moon

**Affiliations:** Department of Biology Education, Daegu University, 201, Daegudae-ro, Gyeongsan-si 38453, Gyeongsangbuk-do, Republic of Korea; domoon@daegu.ac.kr

**Keywords:** curcumin, pyroptosis, NLRP3 inflammasome, Smurf2

## Abstract

Curcumin, a polyphenolic compound derived from *Curcuma longa*, has drawn significant attention for its pleiotropic pharmacological activities, including anti-inflammatory and anticancer effects. Pyroptosis, an inflammatory form of programmed cell death mediated by inflammasome activation and gasdermin cleavage, has emerged as a critical target in both chronic inflammatory diseases and cancer therapy. This review comprehensively explores the dual roles of curcumin in the regulation of NLRP3 inflammasome-mediated pyroptosis. Curcumin exerts inhibitory effects by suppressing NF-κB signaling, attenuating mitochondrial reactive oxygen species (ROS) and ER stress, preventing potassium efflux, and disrupting inflammasome complex assembly. Conversely, in certain cancer contexts, curcumin promotes pyroptosis by stabilizing NLRP3 through the inhibition of Smurf2-mediated ubiquitination. Molecular docking studies support curcumin’s direct binding to several pyroptosis-associated proteins, including NLRP3, AMPK, caspase-1, and Smurf2. These context-dependent regulatory effects underscore the therapeutic potential of curcumin as both an inflammasome suppressor in inflammatory diseases and a pyroptosis inducer in cancer.

## 1. Introduction

Pyroptosis is a lytic and highly inflammatory form of programmed cell death that plays a pivotal role in both innate immunity and disease progression [1]. Unlike apoptosis, which is generally non-inflammatory and immunologically silent, pyroptosis is characterized by cell swelling, membrane rupture, and the release of intracellular contents including damage-associated molecular patterns (DAMPs) and pro-inflammatory cytokines such as interleukin (IL)-1β and IL-18 [2,3]. This process is predominantly executed by gasdermin family proteins (e.g., GSDMD, GSDME), which form membrane pores upon cleavage by activated caspases such as caspase-1 in the canonical inflammasome pathway [4], and caspase-4/5/11 in the non-canonical pathway [5]. Additional pathways involve granzyme-mediated cleavage in immune cells, highlighting the complexity and context-dependent regulation of pyroptotic death.

In the context of cancer, pyroptosis exhibits a paradoxical duality, functioning as both a tumor-suppressive and tumor-promoting mechanism depending on the cellular environment, the specific gasdermin involved, and the intensity and duration of activation [6,7,8,9]. On one hand, chronic low-grade pyroptosis can facilitate tumor development by maintaining a persistent inflammatory microenvironment. This condition supports angiogenesis, tumor cell proliferation, and immune evasion. For example, GSDME-mediated pyroptosis has been shown to promote the development of colitis-associated colorectal cancer by releasing HMGB1, a prototypical DAMP, which in turn activates the ERK1/2 signaling pathway and upregulates proliferating cell nuclear antigen (PCNA), contributing to tumor proliferation [10]. Similarly, asbestos exposure-induced pyroptosis is implicated in the pathogenesis of malignant mesothelioma, highlighting a link between persistent inflammation and tumorigenesis [11].

On the other hand, acute and robust induction of pyroptosis within tumor cells has emerged as a promising anticancer strategy. When activated explosively, pyroptosis results in immunogenic cell death, releasing a surge of DAMPs and pro-inflammatory cytokines that recruit and activate components of the immune system. IL-1β and IL-18 released during pyroptosis enhance the activation and maturation of dendritic cells (DCs), promote CD8^+^ cytotoxic T lymphocyte responses, and recruit natural killer (NK) cells. These immune events contribute to a shift from an immunosuppressive to an immunostimulatory tumor microenvironment, ultimately enhancing antitumor immunity [12,13,14]. Therefore, selectively suppressing chronic inflammation-driven pyroptosis while promoting acute pyroptosis in tumor cells offers a dual-pronged therapeutic strategy that targets both the tumor and its immunological niche.

Among the natural compounds investigated for their ability to modulate pyroptosis, curcumin, a bioactive polyphenol derived from *Curcuma longa*, has gained substantial attention due to its pleiotropic biological activities and excellent safety profile [15,16,17]. Curcumin has been shown to suppress chronic inflammation-related pyroptosis by downregulating the NLRP3 inflammasome, reducing ROS production, and inhibiting the release of IL-1β and IL-18 in various cell types and disease models [18,19]. These anti-inflammatory effects are particularly relevant in preventing the tumor-promoting consequences of prolonged pyroptotic signaling. For instance, in macrophages and endothelial cells, curcumin inhibits NLRP3 activation and reduces pro-inflammatory cytokine secretion, thereby attenuating chronic inflammation in the tumor microenvironment.

In parallel, curcumin can also function as an inducer of pyroptotic cell death in tumor cells. It has been reported to increase the expression and activation of caspase-1 and GSDME in certain cancer cell lines, leading to membrane pore formation, release of immunostimulatory factors, and enhanced recruitment of immune effector cells [20,21]. These effects culminate in both direct tumor cell killing and stimulation of antitumor immune responses. Moreover, curcumin has been found to sensitize tumor cells to other pyroptosis-inducing agents, suggesting its potential use in combination therapy to enhance cancer immunogenicity.

Taken together, these findings suggest that curcumin holds great promise as a context-dependent modulator of pyroptosis. By simultaneously suppressing chronic inflammation-associated pyroptosis and promoting acute immunogenic pyroptosis in cancer cells, curcumin may serve as a valuable therapeutic tool in cancer treatment strategies that aim to leverage the immune system while minimizing inflammation-driven tumor progression.

This review aims to provide a comprehensive synthesis of recent advances in understanding how curcumin modulates pyroptosis in the tumor microenvironment, with a particular focus on its dual role in attenuating chronic inflammatory pyroptosis and promoting acute cancer cell pyroptosis as a potential strategy for anticancer therapy.

## 2. Pyroptosis Signaling Pathways

Pyroptosis is mediated through four distinct but interrelated signaling pathways. These include the canonical inflammasome pathway, which activates caspase-1 in response to pathogen- or damage-associated signals; the non-canonical pathway, where caspase-4/5 (in humans) or caspase-11 (in mice) directly sense cytosolic lipopolysaccharide (LPS); the apoptotic caspase-mediated pathway, in which apoptotic caspases like caspase-3, -8, and -9 cleave gasdermin proteins to switch apoptosis to pyroptosis; and the granzymes-mediated pathway, where immune cells such as NK cells and CTLs deliver granzymes that directly cleave gasdermins in target cells. These pathways collectively illustrate the complexity and flexibility of pyroptosis in host defense and disease regulation.

### 2.1. Canonical Inflammasome Pathway

The canonical inflammasome pathway, first identified among inflammasome mechanisms, is a defense system that detects both pathogenic elements (PAMPs) and endogenous stress signals (DAMPs). It involves the assembly of large protein complexes composed of cytosolic pattern recognition receptors (PRRs), the adaptor protein ASC (apoptosis-associated speck-like protein containing a CARD), and inflammatory caspases such as caspase-1 [22].

Common PRRs that participate in this process include several members of the nucleotide-binding oligomerization domain-like receptor (NLR) family, namely NLRP1, NLRP3, and NLRC4, as well as AIM2 and pyrin [23,24]. Each of these PRRs has a unique domain architecture and sensing mechanism. For example, NLRP1 features an N-terminal pyrin domain (PYD), a nucleotide-binding domain (NOD), a leucine-rich repeat (LRR) region, and a C-terminal caspase recruitment domain (CARD) [25]. The PYD is essential for interaction with ASC, while the NOD contributes to ATP-dependent oligomerization. The LRR acts in autoinhibition and ligand detection, and the CARD domain is involved in the direct recruitment of pro-caspase-1. Activators of NLRP1 include anthrax lethal toxin, muramyl dipeptide, and Toxoplasma gondii components [26].

NLRP3, another key PRR, has a similar PYD-NOD-LRR structure but lacks a CARD domain. It can be activated by a wide variety of stimuli such as microbial infections (bacteria, viruses, fungi), danger signals like ATP, uric acid, and reactive oxygen species (ROS), and other host-derived stressors [27]. For instance, extracellular ATP binds to P2X7 receptors, leading to potassium efflux and triggering the assembly of the NLRP3 inflammasome, ultimately activating caspase-1 and promoting IL-1β secretion [28].

In contrast, NLRC4 carries an N-terminal CARD domain, a central nucleotide-binding domain (NBD), and a C-terminal LRR. It recognizes intracellular bacterial components like flagellin and proteins secreted via the bacterial type III secretion system [29].

AIM2, a non-NLR sensor, comprises a PYD and a HIN-200 domain that binds to double-stranded DNA from microbial origins such as viruses or bacteria [30]. Pyrin, another inflammasome sensor, contains a PYD, two B-boxes, and a SPRY/PRY domain. It is activated upon detection of bacterial-mediated modifications of host Rho GTPases [24].

Once activated, these PRRs recruit pro-caspase-1 either directly via their own CARD domains or indirectly through ASC, forming an inflammasome complex. Within this complex, pro-caspase-1 undergoes autocatalytic activation, which leads to two major downstream events: (1) the maturation of pro-inflammatory cytokines IL-1β and IL-18 and (2) the cleavage of gasdermin D (GSDMD). The N-terminal domain of GSDMD (GSDMD-NT) then translocates to the plasma membrane to form pores, triggering pyroptotic cell death [31].

### 2.2. Non-Canonical Inflammasome Pathway

The non-canonical inflammasome pathway functions independently of the classical inflammasome complex and is mainly activated by Gram-negative bacteria. Extracellular LPS promotes the production of type I interferons, which activate their receptors and induce caspase-11 expression [32,33].

Vacuolar Gram-negative bacteria can release LPS into the cytosol after vacuole rupture, a process mediated by guanylate-binding proteins. Cytosolic LPS directly binds to caspase-11, activating it. Activated caspase-11 cleaves gasdermin D (GSDMD), and the released N-terminal fragment forms membrane pores, leading to pyroptosis [34,35]. In humans, caspase-4 and -5 are activated similarly by intracellular LPS. While these caspases do not process pro-IL-1β or pro-IL-18 directly, GSDMD-mediated K^+^ efflux activates the NLRP3 inflammasome and caspase-1, enabling cytokine maturation [36]. Caspase-11 activation also leads to pannexin-1 cleavage and ATP release. ATP then stimulates P2X7 receptors, enhancing K^+^ efflux and further promoting NLRP3 and caspase-1 activation in bone marrow-derived macrophages [37]. Thus, caspase-11 plays a central role in linking cytosolic LPS sensing to both pyroptosis and cytokine release in the non-canonical inflammasome pathway.

### 2.3. Apoptotic Caspases-Mediated Pathway

Pyroptosis can also be initiated by apoptotic caspases, in addition to inflammatory caspases like caspase-1/4/5/11. Caspase-3, typically involved in apoptosis, can cleave GSDME in GSDME-expressing cells, converting apoptosis into pyroptosis. Chemotherapy agents such as cisplatin utilize this mechanism to induce pyroptotic cell death [38]. Caspase-8 also contributes by cleaving GSDMD, particularly during Yersinia infection [39,40]. Inhibition of TAK1 by Yersinia effector YopJ activates a signaling complex involving FADD, RIPK1, and caspase-8 on the lysosomal Rag-Ragulator platform, promoting pyroptosis [41]. Additionally, caspase-8 can cleave GSDMC to trigger pyroptosis in cancer cells [42], while apoptotic caspases-3, -6, and -7 can cleave GSDMB, releasing its pore-forming N-terminal domain and inducing pyroptosis [43].

### 2.4. Granzymes-Mediated Pathway

Cytotoxic immune cells such as natural killer (NK) cells, cytotoxic T lymphocytes (CTLs), and CAR T cells can induce pyroptosis in cancer cells by delivering granzymes through perforin. These granzymes cleave gasdermin family proteins, triggering membrane pore formation and pyroptotic cell death.

Granzyme A (GZMA), although abundant, shows weak cytotoxicity in vitro unless at high concentrations [44,45].

However, it has been linked to inflammatory responses, including the release of pro-inflammatory cytokines [46,47]. GZMA from CTLs can cleave GSDMB, causing pyroptosis in GSDMB-positive cancer cells [48], though GSDMB is not expressed in all human tissues and is absent in mice.

Granzyme B (GZMB), released by NK cells, cleaves GSDME at the same site as caspase-3, releasing its active N-terminal domain to induce pyroptosis [49]. GZMB acts both directly on GSDME and indirectly via caspase-3, offering an alternative pyroptotic pathway in caspase-resistant, GSDME-expressing cancer cells. Granzymes-mediated pyroptosis can amplify inflammation in the tumor microenvironment, enhancing immune cell recruitment and antitumor immunity. The key molecules involved in pyroptosis and their functional characteristics are summarized in Table 1, while the mechanisms underlying pyroptosis induction are illustrated in Figure 1.

## 3. Pyroptosis as a Therapeutic Strategy in Cancer: Mechanisms, Targets, and Context-Dependent Roles Across Tumor Types

Pyroptosis is emerging as a crucial form of regulated cell death with potential implications in various cancers, especially where traditional therapies are limited by resistance mechanisms. In lung cancer, particularly in non-small-cell lung cancer (NSCLC), conventional chemotherapy shows limited efficacy due to the cancer cells’ ability to evade apoptosis [50,51]. Studies show that agents like simvastatin and polyphyllin VI (PPVI) induce caspase-1-dependent pyroptosis through ROS and inflammasome activation, inhibiting NSCLC progression [52,53]. Additionally, natural compounds such as cucurbitacin B and dasatinib have been shown to activate GSDMD or GSDME, resulting in enhanced pyroptotic and apoptotic death of cancer cells, sometimes involving mitochondrial ROS, STAT3, and NF-κB pathways [54,55]. CD8^+^ T cells also depend on GSDMD for effective immune response against NSCLC [56]. Metformin and piperlongumine analogues act via the AMP-activated protein kinase (AMPK)/SIRT1/NF-κB/caspase-3/GSDME axis to induce pyroptosis [57,58]. Other molecules like EF24 derivatives modulate pyroptosis–apoptosis switching through NF-κB signaling [59], and APE1 inhibition promotes both pyroptosis and apoptosis via p53 upregulation [60,61]. Inhibition of maternal embryonic leucine zipper kinase (MELK) using OTSSP167 also promotes pyroptotic and apoptotic death through FOXM1 and Akt suppression [62,63].

In gastric cancer (GC), the activation of pyroptosis-related genes (PRGs) such as GSDMD and GSDME is linked to favorable prognosis, while agents like famotidine, simvastatin, icariin, and diosbulbin-B trigger NLRP3 or caspase-3-mediated pyroptosis [64,65,66,67].

In hepatocellular carcinoma (HCC), pyroptosis serves as a therapeutic mechanism to counteract drug resistance [68]. PRGs, GSDMD, and GSDME act as potential diagnostic and therapeutic markers [69,70]. Knockdown of NIMA-related kinase 7 (NEK7) activates the NLRP3/caspase-1/GSDMD axis to inhibit tumor progression [71]. Compounds such as euxanthone, alpinumisoflavone, miltirone, and cannabidiol trigger pyroptosis via inflammasome activation and ROS accumulation [72,73,74,75]. Estradiol similarly promotes antitumor effects via NLRP3 activation, FOXO3 signaling, and IL-6/STAT3 suppression [76]. Berberine exerts antitumor effects through both caspase-1-dependent pyroptosis and NF-κB/AMPK-mediated apoptosis [77].

In breast cancer (BC), pyroptosis-related genes correlate with prognosis and therapeutic sensitivity [78]. Compounds such as polydatin, cisplatin, dihydroartemisinin, and Nobiletin induce pyroptosis by targeting JAK/STAT, NLRP3/caspase-1, AIM2/caspase-3, and NF-κB pathways [79,80,81,82]. Other agents like tetraarsenic hexoxide and triclabendazole promote GSDME-mediated pyroptosis to suppress tumor growth [83,84].

Colorectal cancer (CRC) is also responsive to pyroptosis-inducing strategies. Arsenic trioxide and ascorbic acid co-treatment activates inflammasome formation and caspase-1-mediated pyroptosis [85]. Decitabine enhances inflammasome expression and triggers apoptosis via miR-133b induction [86,87]. Together, these findings support pyroptosis as a valuable mechanism across multiple cancer types, highlighting its therapeutic potential and the need for further research to optimize its application.

## 4. Pharmacological Potential and Structural Features of Curcumin

Curcumin (CUR), a lipophilic polyphenol derived from the rhizome of *Curcuma longa* (family Zingiberaceae), represents the principal active component among curcuminoids, which also include demethoxycurcumin and bisdemethoxycurcumin [88]. Historically, curcumin has been employed in traditional Ayurvedic practices for managing ailments such as gastrointestinal inflammation, wounds, sinusitis, and indigestion [89].

Recent research highlights curcumin’s broad pharmacological activities, including anti-inflammatory, pro-oxidant and antioxidant, anticancer, and antibacterial effects [90,91,92]. These effects are mediated through the modulation of multiple intracellular signaling pathways such as NF-κB, MAPK, WNT/β-catenin, Hippo, NOTCH, and Akt/mTOR [93,94,95,96,97]. Additionally, curcumin’s potential as a photosensitizer in photodynamic therapy (PDT) against cancer and infections has been proposed [89].

Structurally, curcumin offers rich opportunities for chemical optimization. Its multiple functional groups, including α,β-unsaturated carbonyls capable of Michael addition with thiol-containing proteins [98], phenolic hydroxyl groups contributing to radical scavenging, and a 1,3-diketone system that chelates metal ions, all play key roles in its biological activity [99]. This structure–activity relationship (SAR) highlights curcumin’s versatility as a pharmacophore, making it a compelling candidate for further derivatization and medicinal chemistry development. The structural features of curcumin are illustrated in Figure 2.

Despite its promising pharmacological profile, curcumin suffers from poor pharmacokinetic properties, including low aqueous solubility, limited gastrointestinal absorption, rapid metabolism, and systemic elimination, which significantly restrict its clinical efficacy. To overcome these limitations, several strategies have been developed. Nanoparticle-based delivery systems, such as liposomes, polymeric micelles, and PLGA-based nanocarriers, have been shown to enhance curcumin’s solubility, stability, and bioavailability. Curcumin analogues, including EF24 and GO-Y030, have been synthetically modified to improve metabolic stability and potency. Additionally, chemical conjugation approaches—such as PEGylation, phospholipid complexation, and peptide coupling—have demonstrated improved pharmacokinetic profiles and targeted delivery. These advancements aim to maximize curcumin’s therapeutic potential and facilitate its translation into clinical applications.

## 5. Inhibition of Inflammatory Pyroptosis by Curcumin: Molecular Mechanisms and Cellular Targets

Curcumin has been widely investigated for its anti-inflammatory potential across various disease models. A significant portion of its therapeutic efficacy is attributed to its ability to suppress inflammasome activation, particularly the NLRP3 inflammasome, which is a key driver of innate immune responses and chronic inflammation. Recent studies have revealed that curcumin exerts its inhibitory effects on inflammasomes through multiple interconnected molecular mechanisms that act both at the transcriptional and post-translational levels.

One of the central actions of curcumin involves the attenuation of mitochondrial reactive oxygen species, which are known to trigger the activation of the NLRP3 inflammasome [100]. In LPS-primed murine macrophages, such as J774A.1 cells and primary peritoneal macrophages, curcumin scavenges mitochondrial ROS [19]. This prevents the dissociation of thioredoxin from thioredoxin-interacting protein and subsequently inhibits the interaction between TXNIP and NLRP3, a critical event in inflammasome assembly. Additionally, in these macrophages, curcumin suppresses nuclear factor-kappa B (NF-κB) signaling by inhibiting the phosphorylation and degradation of IκBα and preventing the nuclear translocation of the p65 subunit [100,101]. This leads to downregulation of pro-inflammatory genes including NLRP3, pro-IL-1β, and pro-IL-18, effectively blocking the priming stage of inflammasome activation.

Curcumin also interferes with inflammasome complex formation by inhibiting the oligomerization of the Apoptosis-associated speck like protein containing a caspase recruitment domain CARD (ASC), which is essential for pro-caspase-1 recruitment [102,103]. Immunofluorescence studies in LPS-primed J774A.1 macrophages revealed a marked reduction in ASC speck formation following curcumin treatment, indicating impaired inflammasome assembly.

Moreover, in macrophages, renal tubular epithelial cells, and colonic epithelial cells, curcumin modulates upstream events necessary for inflammasome activation, such as potassium ion efflux and lysosomal membrane destabilization. It prevents potassium efflux, which is necessary for NLRP3 activation, and stabilizes lysosomes to block cathepsin B release, thereby further impairing inflammasome activation [104,105].

In renal tubular epithelial cells from potassium oxonate-induced hyperuricemic mouse models, curcumin suppressed NLRP3 inflammasome activation and reduced serum uric acid, creatinine, and BUN levels, with therapeutic effects comparable to those of allopurinol [104]. These effects were associated with decreased mRNA expression of NLRP3, ASC, caspase-1, and IL-1β.

In hepatic cell lines such as HepG2 and BRL-3A cells, curcumin inhibited the TXNIP–NLRP3 axis under fructose-induced metabolic stress [106]. This effect was accompanied by the upregulation of miR-200a, a known inhibitor of TXNIP, resulting in reduced hepatic steatosis, improved lipid metabolism, and suppressed infiltration of inflammatory cells in the liver tissue of fructose-fed rats.

In neuroinflammatory conditions, including models of epilepsy, depression, and ischemic stroke, curcumin showed protective effects in neuronal cell lines such as SH-SY5Y cells, as well as in hippocampal neurons and glial cells including microglia and astrocytes. Curcumin suppressed NLRP3 inflammasome activation and reduced expression of IL-1β, caspase-1, and TXNIP. These effects were further linked to the inhibition of endoplasmic reticulum stress, the maintenance of mitochondrial membrane potential, and the modulation of signaling pathways such as AMPK, JAK2/STAT3, and NF-κB [107,108].

In WI38VA13 lung epithelial cells subjected to paraquat-induced acute lung injury, curcumin attenuated oxidative stress and inflammatory responses by reducing TXNIP and NLRP3 expression, restoring Bcl-2/Bax balance, and downregulating Notch1 signaling [109].

Taken together, curcumin inhibits inflammasome activation through a combination of antioxidant, ion-regulating, transcription-modulating, and inflammasome-disrupting mechanisms. These effects have been consistently demonstrated across diverse cell types, including macrophages, epithelial cells (renal, hepatic, pulmonary, colonic), neurons, and glial cells. This underscores curcumin’s broad therapeutic potential in conditions involving aberrant inflammasome activation. Table 2 summarizes key studies that have reported the inhibitory effects of curcumin on NLRP3 inflammasome activation.

## 6. Curcumin-Induced Pyroptosis: Molecular Mechanisms and Therapeutic Strategies in Cancer

While curcumin has demonstrated robust anti-inflammatory activity by suppressing inflammasome activation in various non-cancerous disease contexts, emerging evidence suggests that in cancer cells, curcumin can paradoxically induce pyroptosis—a form of inflammatory cell death—through selective activation of inflammasome-related pathways. This dual role highlights its context-dependent functionality and therapeutic versatility in both inflammatory and neoplastic diseases.

A key mechanism through which curcumin induces pyroptosis is via the activation of inflammasome complexes. In non-small cell lung cancer (NSCLC), curcumin stabilizes and activates the NLRP3 inflammasome by suppressing Smurf2, an E3 ubiquitin ligase that normally facilitates NLRP3 degradation. This stabilization promotes ASC and pro-caspase-1 recruitment, culminating in GSDMD cleavage and pyroptosis [110].

In models of acute myeloid leukemia (AML), curcumin appears to operate through a non-canonical inflammasome pathway by upregulating interferon-stimulated gene ISG3, thereby activating NLRC4, AIM2, and IFI16 inflammasome sensors. This triggers caspase-1 activation and GSDMD-mediated pyroptosis. Restoration of GSDMD expression in resistant leukemia cells re-established curcumin-induced pyroptosis, highlighting its dependency on gasdermin-mediated execution [111].

Nanoplatforms have further enhanced curcumin’s pyroptotic potential. One example is “pharm-dots,” a PLGA-based nanoformulation encapsulating non-emissive curcumin that generates singlet oxygen upon photoactivation. This photodynamically triggered system initiates caspase-3/GSDME signaling and results in the controlled pyroptotic death of cancer cells, enabling precise spatial and temporal targeting in tumor therapy [112]. To clarify the distinction between apoptosis and pyroptosis in the context of GSDME-mediated cell death, recent studies have demonstrated that caspase-3 can cleave GSDME to generate its N-terminal fragment, which forms membrane pores and triggers a pyroptosis-like lytic cell death. Wang et al. provided key experimental evidence including LDH release, membrane rupture, and dependency on GSDME expression, supporting the classification of this process as pyroptosis rather than classical apoptosis [38].

Curcumin has also been integrated into calcium-based nanocarriers such as CUR@CaCO_3_-PArg@HA, which exploit the acidic tumor microenvironment for localized release. Curcumin in this system inhibits Ca^2+^ efflux, amplifies ROS levels, and contributes to mitochondrial dysfunction. Simultaneously, poly-L-arginine induces NO release and endoplasmic reticulum-derived Ca^2+^ mobilization, collectively activating caspase-1 and GSDMD to induce pyroptosis [20].

Another calcium-based nanoinducer, CaZCH, incorporates curcumin, calcium ions, and H_2_O_2_ to induce caspase-3/GSDME-dependent pyroptosis through mitochondrial Ca^2+^ overload and oxidative stress. This system not only eliminates tumor cells but also reprograms tumor-associated macrophages (TAMs) from the immunosuppressive M2 to the pro-inflammatory M1 phenotype, enhancing dendritic cell maturation and CD8^+^ T cell-mediated antitumor immunity [113].

In hepatocellular carcinoma (HepG2) cells, curcumin has been shown to directly trigger pyroptosis through ROS accumulation. This activates caspase-3, leading to GSDME cleavage and pore formation. The involvement of ROS was further confirmed through the use of ROS scavengers, which attenuated the pyroptotic response [21].

Curcumin’s pyroptosis-modulating effects are not limited to cancer. In inflammatory joint disease models, such as IL-1β-treated chondrocytes, curcumin inhibited TRPM2 and NLRP3 activation, thereby suppressing caspase-1-driven pyroptosis and reducing cartilage inflammation—suggesting potential in the treatment of osteoarthritis [114].

Innovative nanostructures have further leveraged curcumin’s functionality. For example, a Ca^2+^-based nanoplatform co-loaded with curcumin and H_2_O_2_ was shown to initiate GSDME cleavage and mitochondrial stress-induced pyroptosis. Beyond tumor killing, this strategy also promoted immunogenic cell death (ICD), enhanced antigen presentation, and boosted T cell-mediated immune responses [115].

In liver cancer, curcumin-loaded PLGA microbubbles have been utilized in sonodynamic therapy. Upon ultrasound and photonic stimulation, these microbubbles generate ROS, concurrently activating apoptosis and pyroptosis in HepG2 cells, showcasing curcumin’s utility in multimodal oncologic strategies [116].

In colorectal cancer (CRC) models, curcumin has consistently demonstrated apoptotic activity. However, its ability to trigger NLRP3 inflammasome-dependent pyroptosis appears to be cell line-specific. While evident in SW480 and HCT116 cells, no such effect was observed in LoVo or HT29, indicating the need to explore alternative inflammasome pathways in CRC [117].

Finally, chemical modification of curcumin has yielded analogues with improved pharmacokinetic and anticancer properties. One such analogue, compound B2, exhibited enhanced stability and potency in lung cancer models by increasing ROS production, inducing ER stress, and activating both apoptosis and pyroptosis pathways [118].

Collectively, these findings highlight curcumin’s dualistic and context-sensitive role in regulating pyroptosis. While it can suppress inappropriate inflammation in degenerative diseases, under tumor-specific stimuli or advanced delivery platforms, curcumin potently activates pyroptosis, contributing to both direct tumor cell death and stimulation of antitumor immunity. This versatility underscores its promise as a precision adjunct in cancer therapeutics and warrants further investigation into optimized delivery strategies and target-specific modulation. Table 3 provides an overview of the mechanisms and therapeutic implications of curcumin-induced pyroptosis in cancer. Figure 3 illustrates the dual roles of curcumin in modulating NLRP3 inflammasome-mediated pyroptosis.

However, as pyroptosis is inherently inflammatory, its therapeutic application in tumors must be carefully balanced against potential risks such as systemic cytokine release or local immune overactivation. Future studies should assess optimal dosing, delivery platforms, and combination strategies to maximize antitumor efficacy while minimizing inflammatory toxicity.

## 7. Molecular Docking Evidence for Curcumin Binding to Pyroptosis-Associated Targets

Molecular docking approaches have provided critical insights into the direct interactions between curcumin and key proteins involved in pyroptosis regulation. These simulations offer structural support for curcumin’s modulatory roles beyond its indirect effects via oxidative stress or transcriptional regulation.

Curcumin was predicted to bind directly to the NACHT domain of NLRP3, a critical ATPase-containing region responsible for inflammasome oligomerization and activation. In docking simulations using the human NLRP3 crystal structure (PDB ID: 7ALV), curcumin occupied the same ATP-binding pocket targeted by the selective NLRP3 inhibitor NP3-146, with a reported binding energy of −6.02 kcal/mol [119]. Within this pocket, curcumin formed hydrogen bonds with Glu369 and Ala228, suggesting a potential to interfere with ATP binding. Interestingly, demethoxycurcumin (DMC), a natural curcumin analog, showed a stronger binding affinity (−6.44 kcal/mol) and formed an additional hydrogen bond with Thr439. These subtle structural differences are consistent with DMC’s slightly enhanced efficacy in suppressing NLRP3 inflammasome activation, implying that curcumin may act by blocking ATPase activity and preventing NLRP3 oligomerization.

In parallel, docking studies have shown that curcumin exhibits strong binding affinity toward caspase-1, a critical executioner protease in the pyroptotic pathway. Using the crystal structure of human caspase-1, curcumin was predicted to bind its catalytic domain with a binding energy of approximately −970.7 kJ/mol [120]. It formed hydrogen bonds with key active site residues, including Cys285, His237, and Ser339. These interactions suggest a potential for curcumin to inhibit caspase-1 enzymatic activity by stabilizing its inactive conformation or obstructing substrate access. Moreover, curcumin was also shown to bind moderately to other inflammasome proteins such as NLRP3 (−850.2 kJ/mol) and ASC (−770.2 kJ/mol). Protein–protein docking models indicated that curcumin disrupts inflammasome assembly by weakening interactions between NLRP3 and caspase-1, thereby interfering with complex formation and downstream IL-1β maturation.

Further docking studies support curcumin’s interaction with AMP-activated protein kinase (AMPK), a metabolic sensor that modulates autophagy and inflammation. Using the crystal structure of AMPKα2 (PDB ID: 4CFH), curcumin was found to stably occupy the nucleotide-binding cleft, with a binding energy of −10.3 kcal/mol [121]. This pose involved hydrogen bonds with Glu100, Arg83, and Val24, consistent with potential allosteric activation. These structural predictions align with experimental data showing that curcumin enhances AMPK phosphorylation and activates downstream autophagy in endothelial cells, indicating that curcumin may engage AMPK both indirectly through redox modulation and directly via molecular interaction.

Finally, in the context of cancer, curcumin has been shown to induce NLRP3-dependent pyroptosis by targeting Smurf2, an E3 ubiquitin ligase that mediates NLRP3 degradation. Molecular docking simulations demonstrated that curcumin binds with high affinity to a hydrophobic pocket within the HECT domain of Smurf2 [110]. Although exact binding energy and residues were not specified, curcumin was predicted to form hydrogen bonds and hydrophobic contacts that impair Smurf2’s ubiquitin ligase function. This inhibition stabilizes NLRP3 protein levels, facilitating inflammasome activation, caspase-1 cleavage, and GSDMD-mediated pyroptosis. These mechanistic insights highlight curcumin’s role in modulating post-translational regulators of pyroptosis in cancer cells. The molecular docking of curcumin with pyroptosis-associated proteins is presented in Figure 4.

## 8. Dual Regulatory Roles of Curcumin in Pyroptosis: A Context-Dependent Mechanistic Perspective

Curcumin functions as a multifaceted modulator of pyroptosis, exhibiting both inhibitory and promotive effects depending on the specific cellular environment and pathological condition. These seemingly opposing actions are attributed to its differential targeting of key molecular pathways involved in inflammasome regulation.

In many inflammatory disease models, curcumin has been shown to suppress NLRP3 inflammasome activation and pyroptotic cell death. This inhibitory effect is mediated through several mechanisms. Curcumin reduces upstream stress signals such as mitochondrial reactive oxygen species and endoplasmic reticulum stress. It also inhibits NF-κB activation, thereby downregulating the transcription of inflammasome-related genes including NLRP3, pro-IL-1β, and pro-IL-18. Additionally, curcumin blocks potassium efflux, a critical ionic signal for NLRP3 activation, and interferes with the assembly of inflammasome components by preventing the oligomerization of ASC and caspase-1. Together, these effects contribute to the attenuation of inflammatory cytokine release and cellular pyroptosis.

In contrast, under certain conditions such as in cancer cells, curcumin has been reported to promote pyroptosis. This pro-pyroptotic activity involves the inhibition of Smurf2, an E3 ubiquitin ligase that facilitates the degradation of NLRP3. Curcumin binding leads to the stabilization of NLRP3, which enhances inflammasome assembly, caspase-1 activation, and the cleavage of gasdermin D. This process ultimately induces pyroptotic cell death and may contribute to curcumin’s anticancer effects by enhancing immunogenic cell clearance.

Overall, the regulatory role of curcumin in pyroptosis depends on the biological context and molecular target. It acts as a suppressor of pyroptosis in inflammatory conditions while functioning as an inducer in cancer settings. Understanding these dual actions provides valuable insight into curcumin’s therapeutic potential and highlights the importance of context-specific targeting in its application.

## 9. Conclusions

Curcumin exhibits dual and context-dependent roles in regulating NLRP3 inflammasome-mediated pyroptosis. Its inhibitory effects are well-documented in chronic inflammatory and metabolic disorders, where it mitigates inflammasome activation by targeting key upstream signals such as ROS, ER stress, and NF-κB. At the same time, curcumin has been shown to enhance pyroptotic cell death in cancer by stabilizing NLRP3 via inhibition of Smurf2, thereby activating the inflammasome complex. Molecular docking analyses provide structural insights into these mechanisms, revealing curcumin’s capacity to directly bind and modulate various inflammasome-related proteins. These findings highlight the versatile therapeutic potential of curcumin and warrant further investigation into its application as a context-specific modulator of pyroptosis in both inflammatory and cancerous conditions. Although numerous studies have reported the therapeutic effects of curcumin in regulating pyroptosis and related disease pathways, it is important to acknowledge that many of these findings are based on in vitro or preclinical models. Therefore, further validation through well-designed clinical studies is essential to confirm the translational relevance of curcumin in pyroptosis modulation. Future in vivo studies using dual models that recapitulate both inflammatory and tumor microenvironments are needed to validate the context-specific effects of curcumin-induced pyroptosis and assess its therapeutic safety and efficacy.

## Figures and Tables

**Figure 1 ijms-26-07590-f001:**
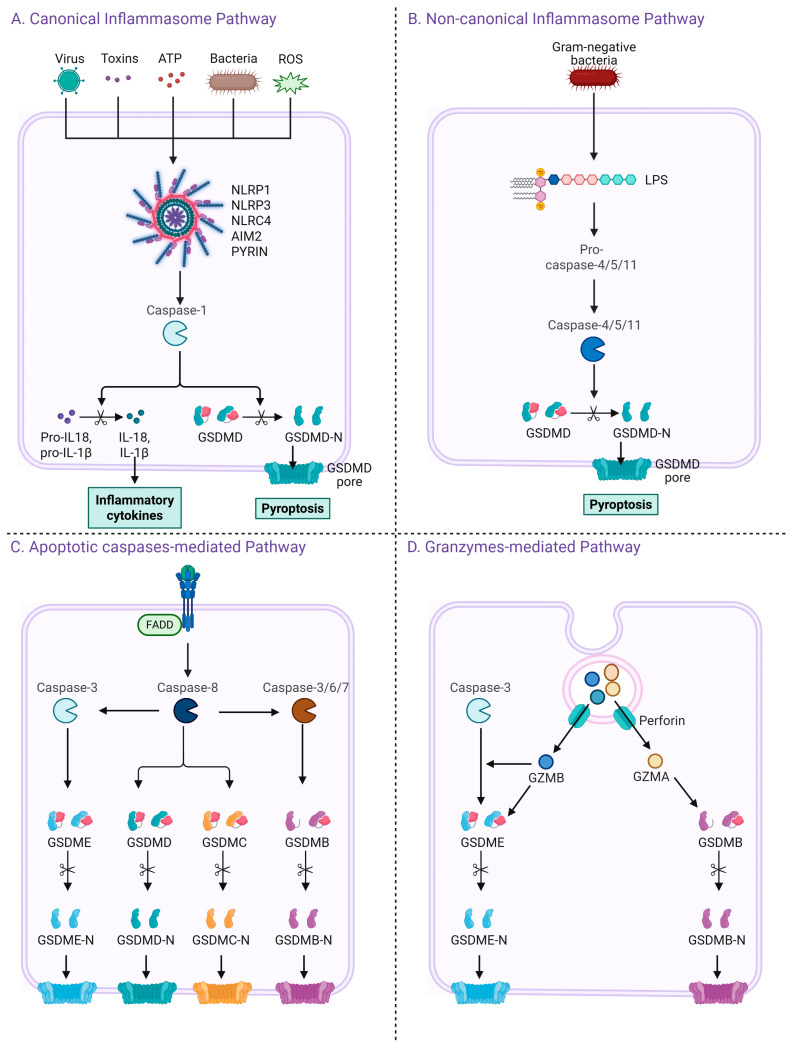
Four distinct pathways leading to pyroptosis and gasdermin activation. (**A**) The canonical inflammasome pathway is triggered by various danger signals including viruses, toxins, ATP, bacteria, and ROS. These stimuli activate cytosolic pattern recognition receptors such as NLRP1, NLRP3, NLRC4, AIM2, and PYRIN, which subsequently activate caspase-1. Activated caspase-1 cleaves GSDMD and pro-inflammatory cytokines pro-IL-1β and pro-IL-18 into their mature forms, leading to pyroptosis and cytokine release. (**B**) The non-canonical inflammasome pathway is activated by intracellular lipopolysaccharide (LPS) from Gram-negative bacteria, which directly activates caspase-4, -5 (human), or caspase-11 (mouse), leading to GSDMD cleavage and pyroptosis. (**C**) In the apoptotic caspase-mediated pathway, caspase-8 or caspase-3/6/7 cleaves several members of the gasdermin family including GSDMD, GSDME, GSDMC, and GSDMB, resulting in the release of their N-terminal domains and formation of membrane pores. (**D**) In the granzyme-mediated pathway, cytotoxic lymphocytes deliver granzymes (GZMA or GZMB) and perforin into target cells. GZMB activates caspase-3, leading to GSDME cleavage, while GZMA directly cleaves GSDMB. Both mechanisms result in pore formation and pyroptotic cell death.

**Figure 2 ijms-26-07590-f002:**
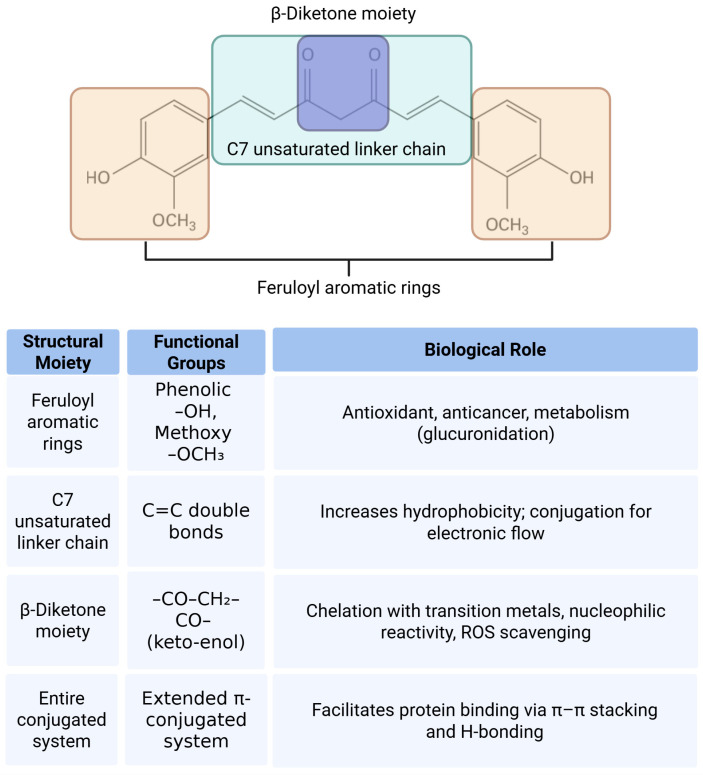
Structural moieties and functional groups of curcumin and their associated biological roles. The diagram illustrates the major structural components of the curcumin molecule including the feruloyl aromatic rings, C7 unsaturated linker chain, and β-diketone moiety. Each region contains specific functional groups such as phenolic hydroxyls, methoxy groups, α,β-unsaturated carbonyls, and a conjugated pi system. These structural features contribute to various biological activities of curcumin including antioxidant, anticancer, metal-chelating, and protein-binding functions. The accompanying table summarizes the relationship between each moiety, its chemical groups, and their respective biological roles.

**Figure 3 ijms-26-07590-f003:**
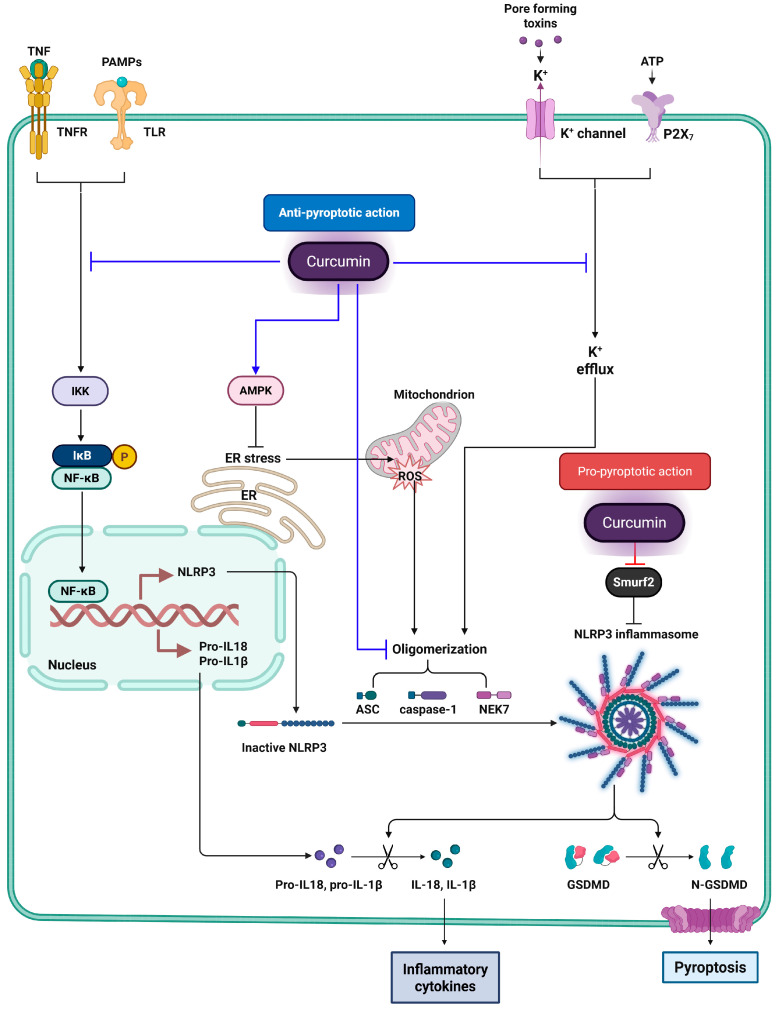
Dual regulatory effects of curcumin on NLRP3 inflammasome-mediated pyroptosis. This figure illustrates the dual modulatory effects of curcumin on NLRP3 inflammasome-mediated pyroptosis. Blue lines represent the anti-pyroptotic actions of curcumin, with flat-headed lines (⊣) indicating inhibition, and arrow-headed lines (→) indicating activation. Curcumin inhibits upstream signaling pathways such as NF-κB by blocking IKK activity, thereby reducing the transcription of NLRP3 and pro-inflammatory cytokines. It also activates AMPK, which alleviates endoplasmic reticulum (ER) stress and decreases mitochondrial ROS production—both of which are known triggers of NLRP3 activation. Additionally, curcumin blocks potassium efflux through inhibition of P2X_7_ receptor and K^+^ channels and disrupts inflammasome assembly by interfering with the oligomerization of ASC and caspase-1. Red lines represent the pro-pyroptotic effects of curcumin. In particular, curcumin inhibits Smurf2, an E3 ubiquitin ligase that promotes NLRP3 degradation. By targeting Smurf2, curcumin stabilizes NLRP3 protein levels, facilitating inflammasome assembly, caspase-1 activation, and GSDMD cleavage, ultimately leading to pyroptotic cell death.

**Figure 4 ijms-26-07590-f004:**
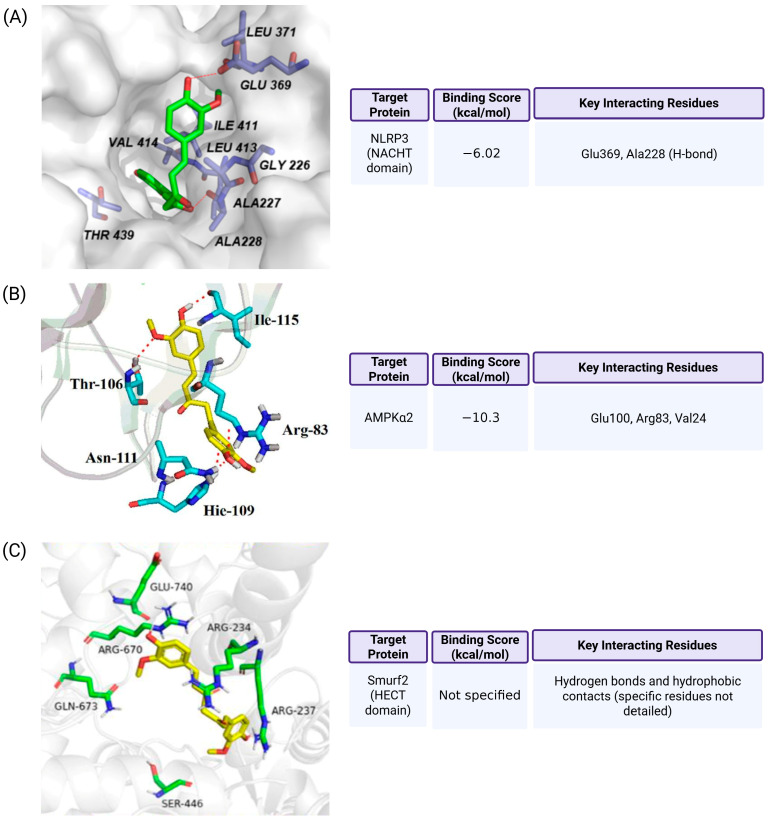
Molecular docking of curcumin with pyroptosis-associated proteins. (**A**) Curcumin binds to the NACHT domain of NLRP3 (PDB ID: 7ALV), overlapping with the binding site of the selective inhibitor NP3-146. It forms hydrogen bonds with Glu369 and Ala228, with a docking score of −6.02 kcal/mol, suggesting moderate to strong affinity (Ref. [119]). (**B**) Curcumin interacts with the nucleotide-binding pocket of AMPKα2 (PDB ID: 4CFH), forming hydrogen bonds with Glu100, Arg83, and Val24. The binding energy is −10.3 kcal/mol, supporting a potential allosteric regulatory role (Ref. [121]). (**C**) Curcumin is predicted to occupy a hydrophobic pocket in the HECT domain of Smurf2, engaging in hydrogen bonding and hydrophobic interactions. Although specific residues and binding score were not provided, this interaction is proposed to inhibit Smurf2’s ubiquitin ligase activity and stabilize NLRP3 (Ref. [110]). Methodological details, including the docking software, receptor preparation, and validation strategies, are available in the cited references (Refs. [110,119,121]).

**Table 1 ijms-26-07590-t001:** Key molecules involved in pyroptosis and their functional characteristics.

Protein Name	Function
GSDMD (Gasdermin D)	Central executor of pyroptosis. Cleaved by caspase-1/4/5/11; the N-terminal fragment forms pores in the plasma membrane, leading to cell lysis.
GSDME (Gasdermin E)	Originally associated with apoptosis; cleaved by caspase-3 to switch cell death from apoptosis to pyroptosis in some contexts.
Caspase-1	Key enzyme in the canonical inflammasome pathway. Activates pro-inflammatory cytokines IL-1β and IL-18 and cleaves GSDMD.
Caspase-4/Caspase-5 (Human)	Recognize intracellular LPS directly and activate the non-canonical inflammasome pathway by cleaving GSDMD.
Caspase-11 (Mouse)	Functional analog of human caspase-4/5 in mice. Activates non-canonical pyroptosis via GSDMD cleavage.
NLRP3	A pattern recognition receptor (PRR) that forms the NLRP3 inflammasome upon activation, recruiting ASC and pro-caspase-1.
NLRC4	Another PRR that forms an inflammasome in response to bacterial flagellin and type III secretion system proteins.
AIM2	Recognizes double-stranded DNA (dsDNA) in the cytoplasm, forming the AIM2 inflammasome.
NLRP1	Sensor that forms an inflammasome in response to various stress signals and pathogen-associated molecules.
IFI16	DNA sensor that can trigger inflammasome formation, particularly in viral infections.
ASC (PYCARD)	Adaptor protein with a CARD domain; bridges inflammasome sensors (e.g., NLRP3) and pro-caspase-1 to facilitate activation.
IL-1β	Pro-inflammatory cytokine activated by caspase-1; promotes fever, inflammation, and immune cell recruitment.
IL-18	Another cytokine activated by caspase-1; enhances NK cell activity and IFN-γ production.
HMGB1	DAMP released during pyroptosis; amplifies inflammation.
NEK7	Serine/threonine kinase that binds NLRP3 to facilitate its activation and inflammasome assembly.
Pannexin-1	Channel protein that may facilitate ATP release during inflammasome activation; associated with pyroptosis initiation.
GBP (Guanylate-binding proteins)	Induced by IFNs; aid in LPS delivery to caspase-11 in the non-canonical pyroptosis pathway.
TLR4	Toll-like receptor that primes inflammasome components via NF-κB pathway activation.
TRIF/MyD88	Adaptor proteins for TLR signaling; regulate transcriptional priming of inflammasome components.

**Table 2 ijms-26-07590-t002:** Key studies demonstrating curcumin’s inhibitory effects on NLRP3 inflammasome activation.

Cell/Animal Model	Key Findings	Mechanistic Highlights	Ref.
THP-1 and RAW264.7 macrophages; MSU-induced gout model in mice	Decreases mRNA and protein levels of IL-1β, IL-6, TNF-α, COX-2, and PGE2 (~2.0–2.5-fold reduction).	Blocks NF-κB priming; protects mitochondria to prevent NLRP3 assembly	[100]
Primary microglia and MCAO stroke mice	Decreases protein levels of IL-1β and IL-18 (~49–52% reduction) and suppresses NLRP3 inflammasome components (NLRP3, ASC, cleaved caspase-1) and GSDMD-N expression.	Suppresses NF-κB pathway; limits NLRP3-mediated microglial pyroptosis	[101]
J774A.1 macrophages (likely LPS + ATP)	Decreases IL-1β secretion and cleaved caspase-1 levels and suppresses NLRP3 inflammasome activation by inhibiting LPS priming, K^+^ efflux, mitochondrial clustering, and ASC speck formation.	Suggested direct inhibition of inflammasome machinery	[102]
Rat liver I/R injury model	Decreases serum and liver levels of TNF-α and IL-6 (by ~19–26% and ~26–36%, respectively), MPO (~32–33%), and NF-κB (~33–46%); increases SOD (~25–39%) and improves histopathological liver injury scores (~39–49%) in intestinal I/R-induced rats.	Prevents inflammasome priming via NF-κB suppression	[103]
Hyperuricemic mice and renal tubular epithelial cells	Decreases serum levels of IL-1β, IL-18, UA, CRE, and BUN and suppresses serum and liver XOD activity, MDA accumulation, and NLRP3 inflammasome activation in kidney; restores SOD and GSH-Px activities in potassium oxonate-induced hyperuricemic mice.	Inhibits priming (NF-κB) and NLRP3 assembly in kidney cells	[104]
LPS-primed macrophages + DSS colitis mice	Decreases IL-1β secretion, caspase-1 activation, IL-6, and MCP-1 levels, while suppressing NLRP3 inflammasome activation (via inhibition of K^+^ efflux, ROS, cathepsin B, and ASC speck formation).	Blocks ROS, K^+^ efflux, and cathepsin B release	[105]
HepG2 and BRL-3A cells; fructose-fed rats	Upregulates miR-200a and downregulates TXNIP and NLRP3 inflammasome activation in fructose-fed rat livers and fructose-exposed BRL-3A/HepG2 cell.	MicroRNA-mediated suppression of TXNIP-NLRP3 signaling	[106]
SH-SY5Y cells; rat hippocampal neurons	Decreases TXNIP expression, NLRP3 and cleaved caspase-1 levels, and IL-1β secretion and suppresses ER stress markers (p-IRE1α, p-PERK) and ROS production via AMPK activation.	Via AMPK-dependent inhibition of ER stress and inflammasome	[107]
Spinal astrocytes in SNI neuropathic pain mice	Decreases spinal mRNA and protein levels of IL-1β and NALP1 (~40–60% reduction) and suppresses cleaved caspase-1, GFAP, and phosphorylated JAK2/STAT3 (~30–55% reduction).	Targets astrocytic inflammasome and neuroinflammatory signaling	[108]
WI38VA13 lung epithelial cells and PQ lung injury rats	Decreases lung mRNA and protein levels of TXNIP, NLRP3, cleaved caspase-1, and IL-1β (~35–60% reduction) and suppresses inflammatory cell infiltration and histopathological lung damage in paraquat-induced acute lung injury model.	Protects epithelial cells by inhibiting TXNIP/NLRP3 activation	[109]

**Table 3 ijms-26-07590-t003:** Mechanisms and therapeutic implications of curcumin-induced pyroptosis in cancer.

Cancer/Model	Curcumin’s Mechanism on Pyroptosis	Ref.
Non-small cell lung cancer	Inhibits Smurf2, stabilizes NLRP3, and promotes NLRP3-mediated pyroptosis via GSDMD; increases NLRP3, cleaved caspase-1, and GSDMD protein levels by ~1.6–2.3 fold (Western blot).	[110]
Acute myeloid leukemia	Increases ISG3 mRNA expression (qPCR, no fold-change reported), activates NLRC4, AIM2, and IFI16 inflammasomes, and increases cleaved caspase-1 and GSDMD-N protein levels and LDH release (Western blot and LDH assay).	[111]
Tumor (general)	Curcumin-loaded PLGA nanoparticles under 660 nm light irradiation generate singlet oxygen, activate caspase-3, cleave GSDME, and induce pyroptotic cell death in tumor cells.	[112]
Tumor (general)	Nano-curcumin disrupts Ca^2+^ homeostasis → activates caspase-1 → GSDMD-N → pyroptosis.	[20]
Colorectal cancer	CaZCH NP releases curcumin and Ca^2+^ → caspase-3 → GSDME-mediated pyroptosis + M2 to M1 TAM switch.	[113]
Hepatocellular carcinoma (HepG2)	Curcumin (30 μM, 12 h) increases ROS production, decreases pro-caspase-3 expression (~50%), upregulates GSDME-N (~2.5-fold), and elevates LDH release (~2.8-fold), leading to pyroptotic cell death.	[21]
Knee osteoarthritis	Modulates TRPM2/NLRP3 signaling (~40% reduction in TRPM2/NLRP3 expression) → suppresses ROS (~39% reduction) → inhibits caspase-1 and GSDMD cleavage → reduces pyroptosis.	[114]
Tumor (general)	Ca^2+^ nanomodulator with curcumin induces mitochondrial Ca^2+^ overload → pyroptosis.	[115]
Liver cancer (in vitro)	Curcumin-loaded microbubbles + SPDT → mitochondrial damage → pyroptosis + apoptosis.	[116]
Colorectal cancer	Upregulates NLRP3 inflammasome components (NLRP3, ASC, caspase-1) (~1.3–1.6-fold, qPCR); partially increases GSDMD expression (~1.2-fold), suggesting partial pyroptosis activation.	[117]
Lung cancer	Curcumin analogue B2 increases ROS levels (~1.8-fold), induces ER stress (↑ p-PERK and CHOP), leading to activation of both apoptosis (↑ cleaved caspase-3, Bax/Bcl-2 ratio) and pyroptosis (↑ GSDMD-N and IL-1β levels) in A549 and H1299 lung cancer cells.	[118]

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
