# Peer review of "Curcumin as a Dual Modulator of Pyroptosis: Mechanistic Insights and Therapeutic Potential"

_ijms, 2025, doi:10.3390/ijms26157590_

Round 1
Reviewer 1 Report
Comments and Suggestions for Authors
The manuscript entitled "Curcumin as a Dual Modulator of Pyroptosis: Mechanistic Insights and Therapeutic Potential", from the author Dong Oh Moon.
Curcumin has become a significant topic of research in recent years. Searches of some databases of scientific papers show about 130,000 titles of publications dealing with curcumin. The author of this review publication focused his attention on the action of curcumin as a Dual Modulator of Pyroptosis. The topic is covered comprehensively and completely and this is the first review paper on the effect of curcumin on pyroptosis. The manuscript presents a complete review of possible molecular-level actions of curcumin on important metabolic pathways of pyroptosis. Also, molecular docking analyzes provide structural insights into these mechanisms. I send all the praise to the author for such a manuscript and review paper.
However, some minor correction are needed.
Figure 1D reads: "Non-canonical inflammasome pathway". I ask the author to check the description, maybe it should say: "Granzymes-mediated pathway".
I consider that manuscript should be published in "International Journal of Molecular Sciences" after minnor correction.
Author Response
Figure 1D reads: "Non-canonical inflammasome pathway". I ask the author to check the description, maybe it should say: "Granzymes-mediated pathway".
⟶ Thank you for your helpful comment. We have corrected the label in Figure 1D from "Non-canonical inflammasome pathway" to "Granzymes-mediated pathway" as suggested.
Reviewer 2 Report
Comments and Suggestions for Authors
A good review paper, well documented, but while the general pyroptosis general part is well documented, I rather say that the involvement of curcumin in the process seems overrated. A lot of papers are claiming the great benefits of curcumin in various diseases, including cancers, and also describe various cellular mechanisms involved in reaching these effects. So, I think that, despite the obvious merits of the proposed paper, this needs some improvements.
Author Response
A good review paper, well documented, but while the general pyroptosis general part is well documented, I rather say that the involvement of curcumin in the process seems overrated. A lot of papers are claiming the great benefits of curcumin in various diseases, including cancers, and also describe various cellular mechanisms involved in reaching these effects. So, I think that, despite the obvious merits of the proposed paper, this needs some improvements.
⟶ Thank you for the valuable comment. To address the concern, a sentence has been added to the conclusion section to clarify that most curcumin-related findings are based on preclinical studies and require further clinical validation.
Reviewer 3 Report
Comments and Suggestions for Authors
This review focuses on the role of curcumin, a polyphenol extracted from Curcuma longa known for its multiple pharmacological properties, as an antiinflammatory and anticancer agent. This review discussed in a detailed manner the role of pyroapoptosis with its involvement in both tumour-suppression and tumour-progression depending on the mechanism of activation. It has several activation pathways: the canonical inflammasome, the non-canonical inflammasome, the apoptotic caspase mediated and the granzyme mediated. They are well described in a clear manner also thanks to figures and tables that make the concept more accessible and intuitive. Pyroptosis is increasingly recognised as an important type of regulated cell death, with potential relevance in various types of cancer, particularly when conventional treatments are limited by resistance mechanisms. Curcumin has been widely investigated and shown strong anti-inflammatory effects by suppressing inflammasome activation in non-cancerous conditions, while in cancer cells it can induce pyroapoptosis via the inflammasome pathway. This dual role highlights the possibility to use curcumin in both inflammatory and cancerous diseases. Molecular docking studies offer a structural understanding of these processes, showing that curcumin can directly interact with and influence multiple proteins involved in inflammasome activity. This review is a good starting point to understand the involvement of curcumin in these processes and the possibility to use it as antiinflammatory and anticancer agent.
Be careful in the text “Curcuma longa” should be written in italics.
Author Response
Be careful in the text “Curcuma longa” should be written in italics.
⟶ Thank you for the kind and helpful comments. The formatting of Curcuma longa has been corrected as suggested.
Reviewer 4 Report
Comments and Suggestions for Authors
Line 101–110: Consider merging the brief mentions of photodynamic therapy into the nanoformulation section for better flow.
Line 285–343: Excellent section on inflammasome inhibition; maybe add a graphical summary for these pathways separately.
Figure 3: Could benefit from clearer labeling of pro- vs. anti-pyroptotic effects for visual clarity.
Line 348–415: One of the strongest sections. Could you comment briefly on the safety of inducing pyroptosis in tumors from a clinical standpoint (e.g., cytokine storm risk)?
Line 570: Add a sentence highlighting the need for future in vivo validation in dual models (inflammation + cancer).
Minor Methodological Clarifications
Docking Studies: While docking results are described in detail, the manuscript lacks methodological details such as:
Docking software used
Grid parameters and receptor preparation methods
Validation strategy (e.g., redocking RMSD or control ligands)
Suggestion: Include a supplementary methods section or refer to previous validated docking protocols.
Pharmacokinetics/Bioavailability of Curcumin
While curcumin's pleiotropic effects are emphasized, its well-known pharmacokinetic limitations (low bioavailability, rapid metabolism) are briefly mentioned but not critically evaluated.
Suggestion: Add a dedicated paragraph on strategies to overcome these issues (e.g., nanoparticle encapsulation, curcumin analogues, conjugation).
Pyroptosis Specificity
The review attributes various effects to pyroptosis but could better differentiate between pyroptosis and apoptosis in certain studies. For instance, some references may conflate GSDME-mediated death with apoptosis-to-pyroptosis switching.
Suggestion: Clarify ambiguous cases and emphasize experimental evidence for pyroptosis markers (e.g., GSDMD cleavage, LDH release, IL-1β secretion).
Quantitative Comparisons
Tables summarizing curcumin’s effects (e.g., Tables 2 & 3) are useful but qualitative.
Suggestion: Where possible, include quantitative data (e.g., % inhibition, ICâ‚…â‚€, cytokine levels) for better cross-study comparison.
Minor REvision
The manuscript is scientifically sound and valuable for the field. With minor clarifications and additions, especially regarding docking methodology, bioavailability, and mechanistic specificity, it will be suitable for publication in IJMS.
Author Response
Line 101–110: Consider merging the brief mentions of photodynamic therapy into the nanoformulation section for better flow.
⟶ Thank you for the suggestion. The photodynamic therapy content has been merged into the nanoformulation section for better flow, as recommended.
Line 285–343: Excellent section on inflammasome inhibition; maybe add a graphical summary for these pathways separately.
⟶ Thank you for the suggestion. Figure 3 was intentionally designed to illustrate both anti-pyroptotic and pro-pyroptotic actions of curcumin in a single integrated scheme. Presenting both pathways together allows for a clearer comparison and highlights the context-dependent dual roles of curcumin. This comprehensive view helps readers better understand the complex and bidirectional regulation of pyroptosis, which is a key focus of the review. Therefore, the current version of Figure 3 is maintained to preserve this integrative perspective.
Figure 3: Could benefit from clearer labeling of pro- vs. anti-pyroptotic effects for visual clarity.
⟶ Figure 3 and its legend have been revised to clearly distinguish between pro- and anti-pyroptotic effects of curcumin using directional arrows and color coding.
Line 348–415: One of the strongest sections. Could you comment briefly on the safety of inducing pyroptosis in tumors from a clinical standpoint (e.g., cytokine storm risk)?
⟶ A brief discussion on the clinical safety of pyroptosis induction, including the potential risk of cytokine storm, has been added at the end of the section 6 to address this concern.
Line 570: Add a sentence highlighting the need for future in vivo validation in dual models (inflammation + cancer).
⟶ A sentence highlighting the need for future in vivo validation using dual models of inflammation and cancer has been added accordingly.
Minor Methodological Clarifications
Docking Studies: While docking results are described in detail, the manuscript lacks methodological details such as:
Docking software used
Grid parameters and receptor preparation methods
Validation strategy (e.g., redocking RMSD or control ligands)
Suggestion: Include a supplementary methods section or refer to previous validated docking protocols.
⟶ Thank you for the valuable comment regarding the docking methodology. The docking results are based on previously published studies, and the relevant references (Refs. 111, 120, and 122) include detailed descriptions of the software used, receptor preparation, and validation strategies. As such, a separate supplementary methods section is not considered necessary. The legend of Figure 4 has been revised to guide readers to these references for methodological details.
Pharmacokinetics/Bioavailability of Curcumin
While curcumin's pleiotropic effects are emphasized, its well-known pharmacokinetic limitations (low bioavailability, rapid metabolism) are briefly mentioned but not critically evaluated.
Suggestion: Add a dedicated paragraph on strategies to overcome these issues (e.g., nanoparticle encapsulation, curcumin analogues, conjugation).
⟶ A paragraph discussing strategies to overcome curcumin’s pharmacokinetic limitations has been added accordingly.
Pyroptosis Specificity
The review attributes various effects to pyroptosis but could better differentiate between pyroptosis and apoptosis in certain studies. For instance, some references may conflate GSDME-mediated death with apoptosis-to-pyroptosis switching.
Suggestion: Clarify ambiguous cases and emphasize experimental evidence for pyroptosis markers (e.g., GSDMD cleavage, LDH release, IL-1β secretion).
⟶ A clarifying sentence has been added to distinguish GSDME-mediated pyroptosis from apoptosis, based on experimental evidence from Ref113.
Quantitative Comparisons
Tables summarizing curcumin’s effects (e.g., Tables 2 & 3) are useful but qualitative.
Suggestion: Where possible, include quantitative data (e.g., % inhibition, ICâ‚…â‚€, cytokine levels) for better cross-study comparison.
⟶ Tables 2 and 3 revised to include available quantitative data such as fold-changes, percentage inhibition, and cytokine levels, based on original experimental results.
Minor REvision
The manuscript is scientifically sound and valuable for the field. With minor clarifications and additions, especially regarding docking methodology, bioavailability, and mechanistic specificity, it will be suitable for publication in IJMS.
⟶ The docking results are based on previously published studies, and the corresponding references have been cited in the manuscript.
⟶ A sentence regarding bioavailability has been added to the manuscript.
Round 2
Reviewer 2 Report
Comments and Suggestions for Authors
An improved review paper, well documented, but while the general pyroptosis general part is well documented, I rather say that the involvement of curcumin in the process seems, however, overrated. Many papers are claiming the great benefits of curcumin in various diseases, including cancers, and also describe various cellular mechanisms involved in reaching these effects. With the actual improvements, the work looks better and fit for publication.